# Hair Microbiome Diversity within and across Primate Species

Catherine Kitrinos,[a] Rachel B. Bell,[b] Brenda J. Bradley,[c,d] Jason M. Kamilar[a,b]

[a]Department of Anthropology, University of Massachusetts, Amherst, Massachusetts, USA
[b]Graduate Program in Organismic and Evolution Biology, University of Massachusetts, Amherst, Massachusetts, USA
[c]Center for the Advanced Study of Human Paleobiology, The George Washington University, Washington, DC, USA
[d]Department of Anthropology, The George Washington University, Washington, DC, USA

**ABSTRACT** Primate hair and skin are substrates upon which social interactions occur and are host-pathogen interfaces. While human hair and skin microbiomes display body site specificity and immunological significance, little is known about the nonhuman primate (NHP) hair microbiome. Here, we collected hair samples ($n = 158$) from 8 body sites across 12 NHP species housed at three zoological institutions in the United States to examine the following: (1) the diversity and composition of the primate hair microbiome and (2) the factors predicting primate hair microbiome diversity and composition. If both environmental and evolutionary factors shape the microbiome, then we expect significant differences in microbiome diversity across host body sites, sexes, institutions, and species. We found our samples contained high abundances of gut-, respiratory-, and environment-associated microbiota. In addition, multiple factors predicted microbiome diversity and composition, although host species identity outweighed sex, body site, and institution as the strongest predictor. Our results suggest that hair microbial communities are affected by both evolutionary and environmental factors and are relatively similar across nonhuman primate body sites, which differs from the human condition. These findings have important implications for understanding the biology and conservation of wild and captive primates and the uniqueness of the human microbiome.

**IMPORTANCE** We created the most comprehensive primate hair and skin data set to date, including data from 12 nonhuman primate species sampled from 8 body regions each. We find that the nonhuman primate hair microbiome is distinct from the human hair and skin microbiomes in that it is relatively uniform—as opposed to distinct—across body regions and is most abundant in gut-, environment-, and respiratory-associated microbiota rather than human skin-associated microbiota. Furthermore, we found that the nonhuman primate hair microbiome varies with host species identity, host sex, host environment, and host body site, with host species identity being the strongest predictor. This result demonstrates that nonhuman primate hair microbiome diversity varies with both evolutionary and environmental factors and within and across primate species. These findings have important implications for understanding the biology and conservation of wild and captive primates and the uniqueness of the human microbiome.

**KEYWORDS** ecology, evolution, integument, mammal, skin

Address correspondence to Catherine Kitrinos, ckitrinos@umass.edu, or Jason M. Kamilar, jkamilar@umass.edu.

The authors declare no conflict of interest.

Microbiome diversity—the "catalog" of host-associated microbial taxa and their collective genes (1)—has important implications for host biology and health (2). Animal microbiome diversity can influence, or be influenced by, metabolism (3), behavior (4), and importantly immunity (5). Both environmental and evolutionary factors shape animal microbiome diversity, including but not limited to habitat (6), captivity status (7), diet (8), social contact (9), birth mode (10, 11), sex (12), and host genetic variation (13). Phylosymbiosis—the apparent correlation between host phylogeny and microbiome diversity—has also been observed in numerous studies

(14–17), suggesting that any factor which varies with host evolutionary history may impact microbiome structure (16).

Most of our knowledge of animal (including human) microbiome diversity comes from studies of the gut (18, 19), and little is known about the diversity of other body regions, such as the hair. Hair is a defining feature of all mammals and plays a critical role in numerous aspects of their biology. Aside from functioning in thermoregulation, hair provides camouflage from predators (20, 21), signaling to conspecifics (20) and—in primates and other social species—is an important substrate through which dominance hierarchies are established and social cohesion is bolstered by grooming. Hair is also home to multiple ectoparasites, including lice, ticks, and mites (22–24), making it an important host-pathogen interface. As ectoparasitic infections may result in conditions, such as anemia, and even death (22, 25), they can incur a substantial cost to fitness in the host. Therefore, symbiotic relationships that control "hair-borne" pathogen spread may confer an adaptive advantage to the host. Indeed, fungal isolates from sloth hair have been shown to display antimicrobial activity as well as activity against parasites (26).

Commensal microbes of the skin (5) and gut (27, 28) influence host immune response. Skin-associated microbial symbionts (*Staphylococcus hominis* and *Staphylococcus epidermis*) act as a first line of defense against pathogenic variants of *Staphylococcus* by targeting them with antimicrobial peptides (AMPs) and working synergistically with the host immune system (5). Hair has a close relationship with the skin, being a skin appendage, with parts of the proximal portion of the hair belonging to the skin environment (29). Hair forms the outermost barrier between host and the environment across many parts of the nonhuman primate body, so hair-associated microbes may play a similar role to skin microbes in host defense.

In humans, skin and hair microbial diversity vary between sexes, individuals (12, 29–31), and especially between body sites (29, 30). Both human and bat hair microbiomes may also be influenced by physical contact or sociality (12, 32). Human hair appears to be rich in human skin-associated microbiota such as *Staphylococcus* and *Corynebacterium* (30), while in a bat microbiome study, oral-associated *Streptococcus salivarius* was the most common species in both the hair and the gut (32). Kolodny et al. also found that temporality and individual identity had different impacts on the bat hair microbiome between open and captive bat colonies and that bacterial abundances correlated with various volatile hair compounds, which can influence scent. These findings suggest that captivity status influences hair microbiome structure and that hair microbes may play an important role in olfactory excretions and therefore social interactions.

A notable aspect of the human hair microbiome is variation across body sites and between sexes. For example, a comparison of the pubis and scalp found *Staphylococcus* was highly abundant in both regions while *Corynebacterium* was differentially abundant in the pubis (40%) and the scalp (7%) (30). In another human hair study, skin-associated bacteria (*Corynebacterium* and *Anaerococcus* spp.) were abundant in the scalp and pubis but *Lactobacillaceae*—a bacterial family found commonly in the human gut and vaginal microbiome—was the most prevalent taxa in female pubic hair. Unlike pubic hair, scalp hair microbial diversity did not differ noticeably with sex, although females did have more "transient" scalp microbiota than males (12). The authors propose that this may be due to more frequent washing and use of products in hair by females than those by males, which results in less stable microbial hair communities (12). However, in the same study, a male-female couple's hair samples clustered more closely (compared to their previous samples) in a principal coordinates analysis when intercourse occurred 18 h prior, despite the fact that the couple showered in the interim (12). Therefore, the evidence for an effect of washing on hair microbiome diversity in humans is mixed, although it has been shown to alter microbiome community composition on the skin of the human hand and to alter it differentially between the sexes (33). Another study found that skin microbial communities—as well as antimicrobial activity—are fairly resilient to normal washing, and thus, even changes to the skin microbiome during normal washing may be temporary (34).

In other mammalian species, differences in scent gland distribution and activity between males and females may be driving sex differences in the microbial diversity of mammalian skin and hair. For example, olfactory signals play an important role in mate choice in many mammalian species, and microbes may modulate scent profiles by adding volatile metabolites (35). Sex differences in scent gland microbiomes have been observed in lemurs (19), bats (35), and hyenas (36). In wild spotted hyenas, juvenile males harbor more taxonomically rich scent gland microbiomes than juvenile females, which may be due to more frequent scent marking in male hyenas than that in females (36). Lemurs rely heavily on olfaction compared with other primates, and differences in prevalent microbial taxa have been noted between the sexes and between dominant and nondominant males (19). Thus, there may be differences in hair microbiome diversity between sexes due to differences in olfactory signaling.

The hair microhabitat lies adjacent to the skin. Additionally, sections of the hair—such as the hair follicle—are part of the skin environment (29). Microbial diversity on hair shafts has been shown to resemble that of the cutaneous skin region from which the shaft originated (30). Microbiome diversity of human hair and skin varies substantially across body sites and microhabitats, with sebaceous sites (oily skin sites with lipid-rich sebum secretions) being the most distinct compositionally from the others, such as "dry" sites (skin sites with little moisture) or "moist" sites (skin sites with high moisture: these generally contain more sweat glands) (31, 37). Topographical features, such as sweat glands, play an important role in shaping the skin and subsequently hair microhabitat, as sweat glands contain antimicrobial substances that may prevent colonization by some microbial taxa (37). Thus, we may expect to find that differences in hair microbial diversity correspond with higher-level taxonomic groupings in the order primates (e.g., the parvorders Catarrhini and Plathyrrhini, and suborder Strepsirhini). This is because, primates have both apocrine sweat glands—which are generally nonthermoregulatory, distributed across the whole of the primate body, and are associated with hair follicles and sebaceous glands—and eccrine sweat glands—which are associated with thermoregulatory sweating and vary in abundance and distribution between major primate clades (38). Eccrine glands especially are distributed differentially across the major primate clades, with a low ratio of eccrine to apocrine glands in platyrrhines and strepsirrhines, a higher ratio in many catarrhines (near 1:1), and the highest ratios in apes with humans having nearly 100% of the body surface covered in eccrine glands (38). Supporting this idea, Council et al. (18) found that axillary skin microbiome diversity corresponded with evolutionary distance among humans, chimpanzees, gorillas, macaques, and baboons. Humans also displayed high abundances of *Staphyloccocaceae* while more phylogenetically distant species from humans (baboons and macaques) had increased amounts of microbiota associated with soil, gut, and oral microbial communities (18).

Given these initial studies suggesting that the hair microbiome could play an important role in primate immunity, social signaling, and various physiological functions, understanding how host factors affect the hair microbiome is essential. The goal of the present study was 2-fold, as follows: (i) to characterize hair microbiome diversity within and across 12 primate species and (ii) to identify the factors that explain this variation. We made several predictions regarding the factors explaining microbiome diversity within and across species. First, we predicted that both the evolutionary history of the species and their environment would explain variation in microbiome diversity. Second, we expected that microbiota from different body regions would exhibit distinct patterns, although not to the extent found in humans since nonhuman primates have more uniform hair and skin characteristics across their body. Finally, we predicted that microbiome diversity would differ across sexes, similar to the human condition (12).

## RESULTS

**Relative abundances of microbial taxa.** All hair samples were dominated by species from the *Bacteria* domain, with only five samples containing more than 5% relative abundance of *Archaea*. The most abundant phyla (>15%) across all samples were

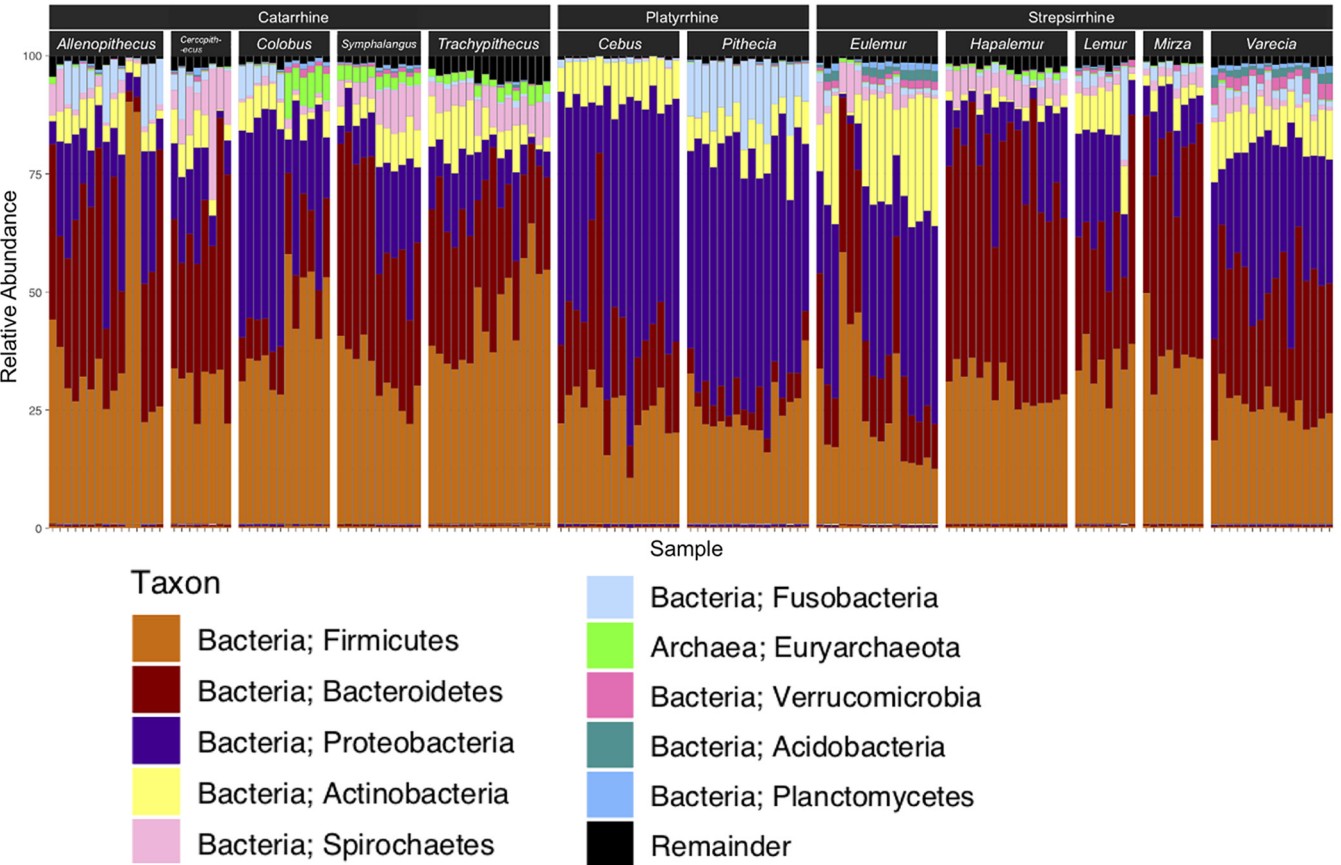

**FIG 1** Relative abundances (%) of the top 10 microbial phyla present in host samples, organized by host species. Each bar represents a sample. The "remainder" category is the aggregate abundance of microbial phyla that were not in the top 10 most abundant group.

*Firmicutes* (32.51%), *Bacteroidetes* (25.42%), and *Proteobacteria* (24.54%) (Fig. 1). The most abundant genera (>1.0%) across samples were bacteria typically associated with the gut, respiratory system, skin, and abiotic environment, such as *Prevotella 9* (8.45%), *Streptococcus* (4.07%), *Treponema* 2 (3.13%), *Prevotella 1* (2.99%), and *Staphylococcus* (2.90%) (see Table S2 and Fig. S1 in the supplemental material).

**Univariate analyses examining hair microbiome diversity.** We found significant differences across species for all measures of alpha diversity (Table 1 and 2; Fig. 2a). We found the highest Chao1 estimates and amplicon sequence variant (ASV) counts in *Lemur catta* followed by *Varecia rubra* and *Eulemur flavifrons*, while the lowest scores were in *Pithecia pithecia* followed by *Cebus capucinus* (which recently has been renamed to *Cebus imitator*) and *Hapalemur griseus* (additional analyses of ASV counts were not performed because of its redundancy with Chao1 results). The highest Shannon diversity values were found in *V. rubra* (H′ = 7.96), *L. catta* (H′ = 7.87), and *Trachypithecus obscurus* (H′ = 7.85), and the lowest score was in *Pithecia pithecia* (H′ = 5.78) (Table 1). Species differences in Pielou's evenness were statistically significant (Kruskal-Wallis, $P < 0.0001$, H = 65.78) with the highest value found in *T. obscurus* (J = 0.80) and the lowest found in *Allenopithecus nigroviridis* (J = 0.68). Based on Faith's phylogenetic diversity, the most phylogenetically diverse host hair samples were from *V. rubra* (84.5) and the lowest were from *P. pithecia* (33.7) (Table 1; Fig. 2a). Using *post hoc* analyses, we found statistically significant differences among most species pairs for each alpha diversity metric (see Table S3 in the supplemental material).

Similarly, we found significant differences across species in weighted and unweighted UniFrac distances (Fig. 3a and b) (see Table S4 in the supplemental material). The principal-coordinate analysis (PCoA) based on unweighted UniFrac distances showed that platyrrhines

**TABLE 1** Species averages for 5 alpha diversity metrics

| Species | Species avg for: | | | | |
|---|---|---|---|---|---|
| | Chao 1 | Shannon | Faith PD | Pielou's | Observed ASVs |
| *Allenopithecus nigroviridis* | 472 | 6.02 | 41.9 | 0.68 | 462 |
| *Cebus capucinus* | 459 | 6.04 | 34.2 | 0.70 | 420 |
| *Cercopithecus neglectus* | 985 | 7.42 | 68.0 | 0.76 | 898 |
| *Colobus angolensis* | 849 | 7.31 | 55.8 | 0.76 | 775 |
| *Eulemur flavifrons* | 1,026 | 7.51 | 72.9 | 0.77 | 945 |
| *Hapalemur griseus* | 461 | 6.40 | 39.2 | 0.73 | 444 |
| *Lemur catta* | 1,364 | 7.87 | 82.8 | 0.77 | 1,215 |
| *Mirza coquereli* | 485 | 6.85 | 36.4 | 0.78 | 468 |
| *Pithecia pithecia* | 354 | 5.78 | 33.7 | 0.70 | 342 |
| *Symphalangus syndactylus* | 694 | 7.41 | 57.9 | 0.79 | 658 |
| *Trachypithecus obscurus* | 1,018 | 7.85 | 63.0 | 0.80 | 918 |
| *Varecia rubra* | 1,288 | 7.96 | 84.5 | 0.78 | 1,183 |

*C. capucinus* and *P. pithecia* clustered in the bottom left of the plot while catarrhines *A. nigroviridis*, *Colobus angolensis*, *T. obscurus*, and *Symphalangus syndactylus* grouped in the center of the plot, and strepsirrhines cluster at either the top left or far right.

We found some sex differences in alpha diversity metrics. In particular, Faith's phylogenetic diversity (PD) differed between males and females ($P = 0.046$, $H = 3.97$) (Fig. 2b) and Chao1 diversity approached statistical significance ($P = 0.055$, $H = 3.67$). In both cases, males had higher values than females. In contrast, we did not find sex differences in Shannon diversity or Pielou's evenness. In addition, we found that sexes exhibited significantly different microbial compositions based on permutational multivariate analysis of variance (PERMANOVA) tests of weighted ($P = 0.01$, pseudo-F = 3.70) and unweighted ($P = 0.001$, pseudo-F = 3.01) UniFrac distances (see Table S5 in the supplemental material).

We did not find significant differences across body sites for any of the alpha or beta diversity metrics (Table 2 and Table S5).

Finally, we found that some alpha diversity metrics significantly varied across the three institutions (Chao1) but others did not. Although, we did find significant differences across institutions using PERMANOVA tests of weighted and unweighted UniFrac distances (Fig. 3c and Table S5). In addition, our *post hoc* analyses showed that each pair of institutions was significantly different for the same beta diversity metrics (Table S5).

**Linear models (LMs) predicting hair microbiome diversity.** We found that species identity was included in the best models predicting each of the eight alpha and beta diversity metrics and was either the most important predictor of each variable (3 out of 8) or shared the highest sum of corrected Akaike's information criterion (AICc) weight values with body site (2 out of 8) or sex (3 out of 8) (Table 3). Sex was a strong predictor of Chao1 and Faith's PD, a moderate predictor of Shannon Diversity, and a weak predictor of Pielou's diversity. In addition, sex was an important predictor for PCoA axis 1 and 2 based on weighted UniFrac distance and PCoA axis 1 based on unweighted UniFrac distance. We found institution to be a moderate predictor of our dependent variables (sum of AICc weight, 0.50), only ranking second in relative impor-

**TABLE 2** Results of univariate analyses[a]

| Dependent variable | Results (H [*P* value]) by independent variable | | | |
|---|---|---|---|---|
| | Species identity | Institution | Sex | Body site |
| Chao 1 | 111.5 (<0.0001) | 6.4 (0.04) | 3.7 (0.056) | 4.9 (0.56) |
| Shannon | 97.7 (<0.0001) | 4.6 (0.10) | 0.2 (0.63) | 7.0 (0.32) |
| Faith PD | 99.7 (<0.0001) | 5.9 (0.051) | 4.0 (0.046) | 8.7 (0.19) |
| Pielou's | 65.8 (<0.0001) | 2.7 (0.25) | 2.1 (0.15) | 5.4 (0.49) |

[a]Differences among species, institutions, sexes, and body sites for alpha diversity metrics determined using Kruskal-Wallis.

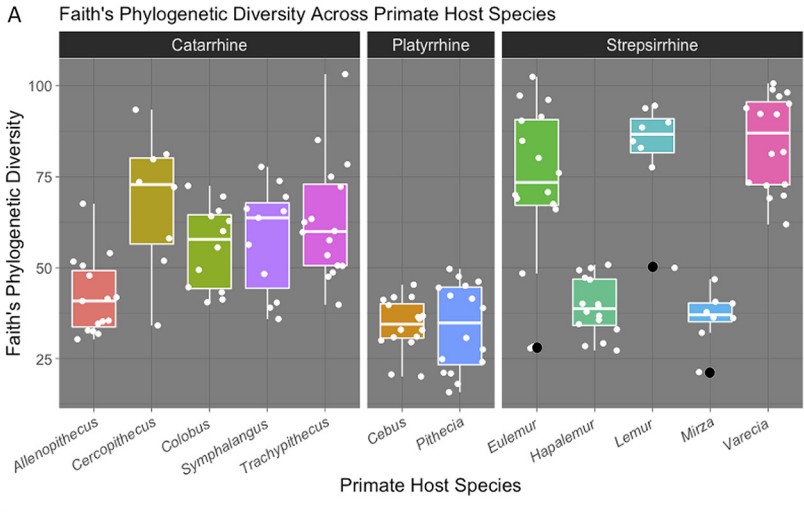

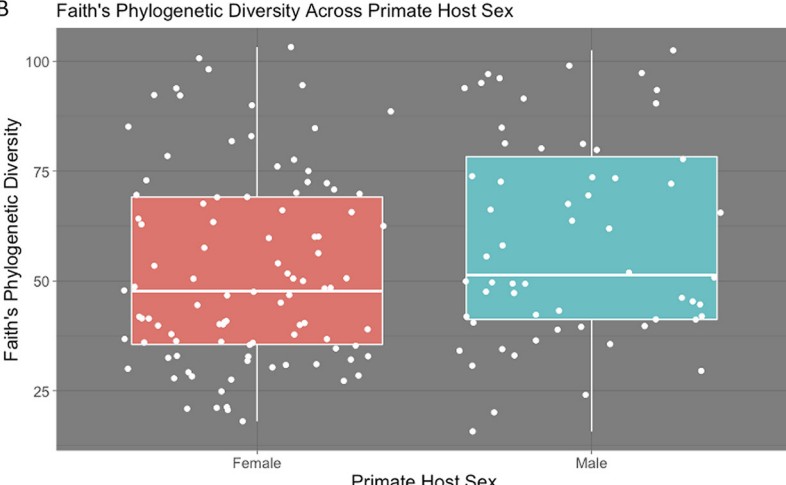

**FIG 2** (a) Boxplot of Faith's phylogenetic diversity across primate host species (H = 99.7, P < 0.0001). White dots represent individual samples. (b) Boxplot displaying Faith's phylogenetic diversity data distribution for male hair samples (n = 60) and female hair samples (n = 98) (H = 4.0, P = 0.046). White dots represent individual samples.

tance as a predictor for Pielou's evenness. For all other dependent variables, institution was the least or second to least important predictor.

In contrast to our univariate analyses, body site was often an important variable explaining alpha and beta diversity metrics in our linear models. Body site was the second most frequently occurring predictor variable in our best models (Table 3). It was ranked with species identity as the most important predictors for Faith's PD, PC1 (weighted UniFrac distances), and PC2 (unweighted UniFrac distances). Also, body site was ranked only slightly below species identity and sex as the most important predictor of Chao1 and PC1 (unweighted UniFrac distances) (sum of AICc weights, 0.92 and 0.99, respectively). Although, *post hoc* analyses showed that this effect was due largely to the difference between the tail—and to a lesser extent the crown and thigh—and other body sites (see Table S6 and S7 in the supplemental material).

## DISCUSSION

We found that primate hair microbiome diversity is best explained by several factors, including their local environment, species identity, sex, and the body site where the sample was obtained. Notably, however, there is less variation in microbiome diversity across body sites compared with that of humans. The more homogenous

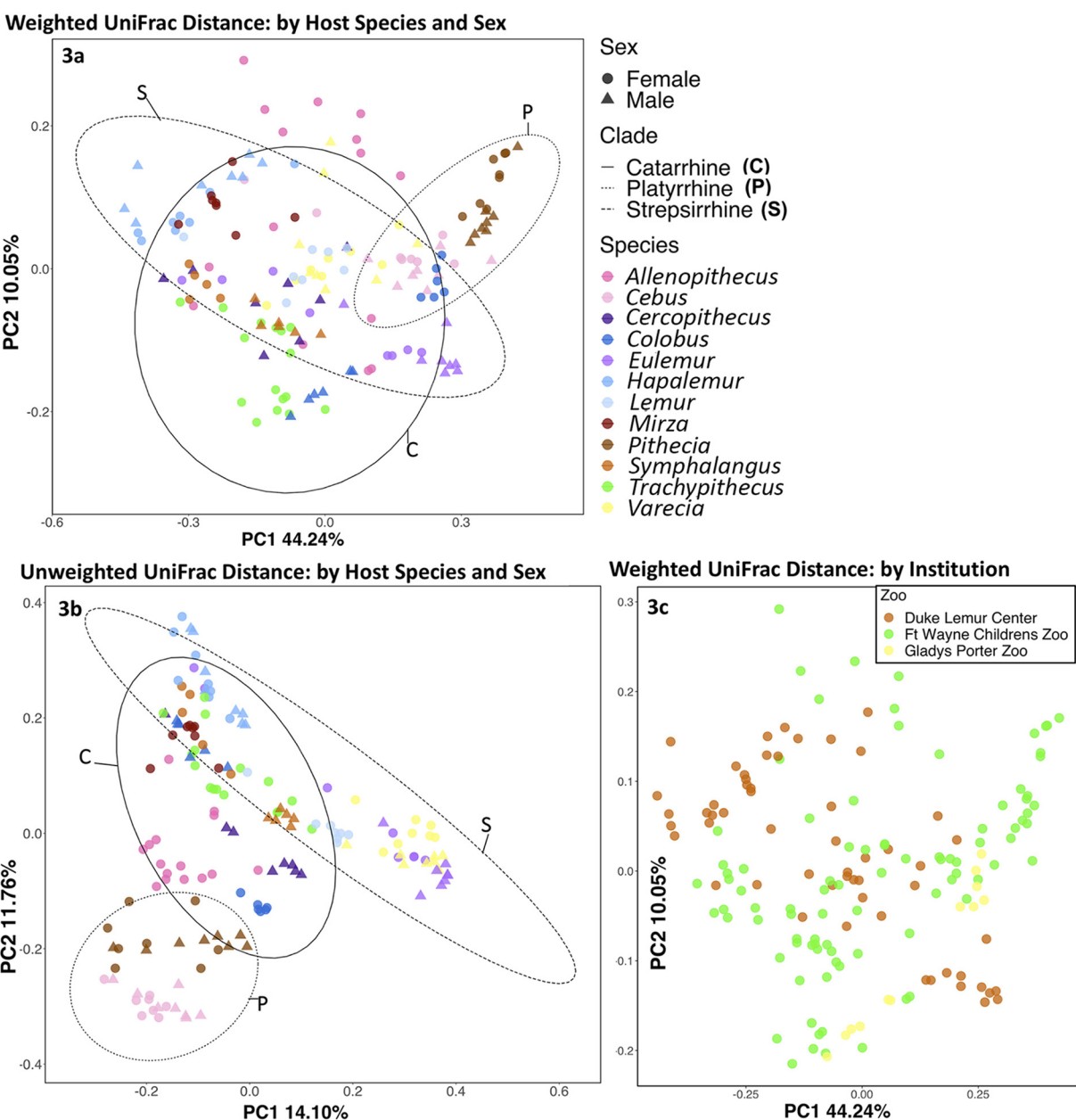

**FIG 3** (a) Principal-coordinate analysis based on weighted UniFrac distances. Each symbol represents a sample. There are significant differences across species based on PERMANOVA ($F = 25.1$, $P = 0.001$). We also identified sex differences (PERMANOVA, $F = 3.7$ $P = 0.014$). Ellipses indicate a 95% confidence interval. The solid line encircles samples from catarrhines, the dotted line encircles samples from platyrrhines, and the dashed line encircles samples from strepsirrhines. (b) Principal-coordinate analysis based on unweighted UniFrac distances. There are significant differences across species based on PERMANOVA ($F = 15.1$, $P = 0.001$). We also found sex differences (PERMANOVA, $F = 3.0$, $P = 0.001$). (c) Principal-coordinate analysis based on weighted UniFrac distances. There are significant differences across institutions based on a PERMANOVA of weighted ($F = 10.0$, $P = 0.001$) and unweighted UniFrac distances ($F = 11.2$, $P = 0.001$).

pattern of hair microbiome diversity across body sites in nonhuman primates may be attributed to the relatively uniform distribution of hair and other skin appendages across their body compared with that of humans (38). Our results are a first step in examining broad patterns in the ecology and evolution of primate hair microbiota and illustrate the unique biology of modern human hair and skin microbiota compared with that of other primates and mammals (17, 18, 29).

**Relative microbial abundances in primate hair.** We found that the most prevalent taxa in our hair samples are reminiscent of the relative abundances found in the human and nonhuman primate (NHP) gut rather than the skin. While our four most abundant

**TABLE 3** Results of linear models predicting hair microbiome alpha and beta diversity metrics[a]

| Dependent variable | Model | AICc | Predictors | | | | Sum of AICc weights | | | |
|---|---|---|---|---|---|---|---|---|---|---|
| | | | Species | Sex | Body site | Institution | Species | Sex | Body site | Institution |
| Chao1 | 1 | 2185.2 | + | + | + | | 1.00 | 1.00 | 0.92 | 0.50 |
| | 2 | 2185.2 | + | + | + | + | | | | |
| Shannon | 1 | 391.9 | + | + | + | | 1.00 | 0.54 | 0.53 | 0.50 |
| | 2 | 391.9 | + | + | + | + | | | | |
| | 3 | 392.3 | + | + | | | | | | |
| | 4 | 392.3 | + | + | | + | | | | |
| | 5 | 392.3 | + | | + | | | | | |
| | 6 | 392.3 | + | | + | + | | | | |
| | 7 | 392.4 | + | | | | | | | |
| | 8 | 392.4 | + | | | + | | | | |
| Faith's PD | 1 | 1269.1 | + | + | + | | 1.00 | 0.99 | 1.00 | 0.50 |
| | 2 | 1269.1 | + | + | + | + | | | | |
| Pielou's | 1 | −375.7 | + | | | | 1.00 | 0.23 | 0.04 | 0.50 |
| | 2 | −375.7 | + | | | + | | | | |
| Weighted PC1 | 1 | −237.4 | + | + | + | | 1.00 | 0.83 | 1.00 | 0.50 |
| | 2 | −237.4 | + | + | + | + | | | | |
| Weighted PC2 | 1 | −392.5 | + | + | | | 1.00 | 0.86 | <0.01 | 0.50 |
| | 2 | −392.5 | + | + | | + | | | | |
| Unweighted PC1 | 1 | −332.5 | + | + | + | | 1.00 | 1.00 | 0.99 | 0.50 |
| | 2 | −332.5 | + | + | + | + | | | | |
| Unweighted PC2 | 1 | −383.6 | + | | + | | 1.00 | 0.39 | 1.00 | 0.50 |
| | 2 | −383.6 | + | | + | + | | | | |
| | 3 | −382.7 | + | + | + | | | | | |
| | 4 | −382.7 | + | + | + | + | | | | |

[a]AICc values for the best models (the model with the lowest AICc value and those within 2 values of this model) predicting each dependent variable are included. All possible predictors are listed and their inclusion in each model is indicated by a "+." The relative importance of each predictor variable for explaining each dependent variable is based on the sum of AICc weights across models, which varies from zero to one (from least to most important).

phyla—*Firmicutes*, *Bacteroidetes*, *Proteobacteria*, and *Actinobacteria*—are also the four commonly found phyla known from a limited data set for the NHP axillary skin (18), the higher prevalence of *Firmicutes* and *Bacteroidetes* in relation to *Actinobacteria* is often associated with gut microbiomes (29). Clayton et al. (2) showed that 22 out of 34 NHP gut microbiome studies had *Firmicutes* and *Bacteroidetes* listed as the most abundant and second most abundant phyla, respectively. Likewise, the most abundant genus in our samples, *Prevotella*, is an especially prominent member of the human gut microbiome—although, it is also present in the human oral microbiome (39)—and has been shown to be particularly prevalent in the "humanized" gut microbiomes of captive primates whose diets are far less diverse than their wild counterparts (6, 7). *Prevotella* and *Bacteroides* have been prevalent in gut microbiome studies of some wild primates, such as wild lorises (40), *Rhinopithecus* (41), and chimpanzees (42). However, in two other studies, Amato and colleagues found that captive black howler monkeys and captive Asian colobines harbored relatively higher abundances of *Prevotella* (6, 43) than their wild counterparts. In an NHP axillary skin microbiome study (18), the primate host species with high abundances of *Prevotella* were either born in captivity (baboons) or were outdoor-living and given rations of "monkey chow" (macaques—who had the highest abundances of *Prevotella*). However, these results may also reflect evolutionary or biological differences between primate host species, as the non-ape primates in the study (baboons and macaques) had proportionally more reads of *Prevotella* and proportionally fewer reads of skin-associated microbial taxa than the apes in the study (18). Therefore, we suggest that our captive primate hair microbiome samples may reflect to some degree a humanized gut microbiome that is somehow impacting the

hair microbiome through transmission of gut microbes to hair or simply the interconnectivity of the gut and skin (44, 45). The potential to be colonized by *Prevotella*, regardless of the microbe's origin, may differ due to differences in host biology occurring at higher clade levels. Future studies that include hair and skin microbiome samples from wild populations will clarify whether or not our results are impacted by captivity.

The high abundance of *Streptococcus* in our primate hair samples is similar to the findings of Kolodny et al. (32), where *Streptococcus salivarius* was the most prevalent species in both the bat hair and gut microbiome. While *Streptococcus* is often associated with the respiratory tract, the fact that other studies have found it commonly in the gut may demonstrate that gut-associated taxa are introduced to the hair through contact with fecal matter and/or that respiratory-associated taxa are introduced to the gut and hair through social contact, particularly social grooming (32). *Treponema* is a generally nonpathogenic genus found commonly in the oral cavity and intestines and is one of the most common genera in the gut microbiomes of NHPs (46). Gut-associated *Treponema* appears to be host specific in primates, and closely related *Treponema* species may appear only on certain host species, suggesting that this genus has colonized primates throughout their evolutionary history but is impacted by lifestyle or diet modifications as it appears to be under negative selection in human populations with "grocery store" diets (46).

**Factors explaining primate hair microbiome diversity. (i) Host species identity.** Similar to studies of gut microbiome diversity, we found that variation in the primate hair microbiome is explained by several factors. Host species identity was consistently the most important predictor of hair microbiome diversity, which may be related to the evolutionary history and unique biological characteristics of the host species. We found noticeable differences among the microbiomes of catarrhines, platyrrhines, and strepsirrhines based on unweighted UniFrac and Bray Curtis distances (Fig. 3b; see Fig. S2 in the supplemental material). Compositional differences in the microbiomes of these primate clades may be due to known variation in their eccrine gland abundance and distributions, which may impact microbe colonization (37, 38). Specifically, eccrine glands in platyrrhines and strepsirrhines are found only on the surfaces of the hands, feet, and (in platyrrhines) the prehensile tail, while in catarrhines, these glands are distributed all over the body (38). These eccrine sweat glands may provide a moist microhabitat that make colonization possible for a more diverse array of microbes. This idea would explain the generally higher alpha diversity in our catarrhines than that in the platyrrhines and some strepsirrhines. Interestingly, these clades also differ in their reliance on olfaction for social communication, with strepsirrhines being the most reliant on scent. Strepsirrhines have scent glands that are not present in monkeys and apes, and some lemurs are capable of emitting hundreds of chemical compounds. For example, the highly social *Lemur catta* is known to produce hundreds of chemical compounds in contrast to *Eulemur*, which secreted only 27 chemical compounds in one study (47). Thus, the very high alpha diversity values for some strepsirrhines could be driven by scent gland deployment, although behavioral data would be necessary to confirm this hypothesis. It has been confirmed in other mammals that integumentary-associated microbial communities vary with scent gland activity (36) and volatile compounds that play a role in olfactory signaling (32). Therefore, hair microbiome structure in primates may be linked to the morphology and activity of other parts of the integument.

In addition to biological variation, our species identity variable may simply reflect the specific social and abiotic environment where each species was housed at their respective institution. However, we do not have data about these environmental characteristics or behavior, so further study is warranted.

**(ii) Host sex.** Host sex was a strong predictor of several alpha and beta diversity metrics, with males scoring higher for measures of taxonomic richness, such as Faith's PD and Chao1. Factors such as hormone cycling and sex-biased microbial transmission may play a role in differentiating male and female NHP hair microbial diversity. Hormone cycling can impact both host olfactory secretions (19) and gut microbiome structure (48), which is especially important considering the high abundances of gut-

associated microbes in our samples. The differential transmission of microbes between the sexes has also been documented in black howler monkeys (*Alouatta pigra*) (49) and in marmosets (50).

In human hair microbiomes, sex differences in microbial diversity are body site specific; females have more transient microbes in their scalp hair than males but fewer transient microbes in the pubic hair than males (12). The higher relative "stability" of female pubic hairs (as well as the high abundances of *Lactobacillus* spp.) is attributed to its proximity to the vaginal microbiome which is dominated by potentially protective *Lactobacillus* spp. (12). However, human females appear to be unique among primates in their high abundances of vaginal *Lactobacillus* spp. (51). Therefore, body site-specific sex differences in nonhuman primate hair microbial communities that arise from differences in reproductive organ ecosystems may be driven by different microbial taxa than those which drive differences in humans, and more research is warranted on this topic.

**(iii) Host institution.** The institution where the primates were housed had a mixed effect on microbiome diversity. Our clearest result was connected to the beta diversity metrics, with primates living in different institutions exhibiting a distinct composition of microbes. The institution functions as the species' environmental context, including their abiotic and dietary characteristics. Although we do not have specific information related to these characteristics for each species, we can assume that institutions differ to some extent. Habitat-dependent variation has been associated with differences in gut microbiome diversity in captive versus wild primates (6). Howler monkeys, for example, living in mostly pristine environments had more varied diets, while those living in fragmented or captive environments had less diverse diets (6). The reduction in diet diversity appears to result in a humanized gut microbiome in nonhuman primates (7), and thus, the high abundances of human gut-associated microbes like *Prevotella* in our samples may signal a dysbiotic gut. Therefore, differences in diet content or variation across institutions may drive differences in the microbiomes of our study species. Exposure to conspecifics creates opportunities for horizontal microbial transmission which can increase overall community diversity and subsequently host health and community resilience (9). Because microbial transmission can also be sex biased (49, 50), the male-female ratios present in the zoo enclosures may impact the hair microbiome structure of the host. Finally, microbes of the abiotic environment are affected by factors such as temperature, UV radiation, and atmospheric carbon dioxide concentration that vary across geographic regions (52). Because primates and other organisms interact constantly with their abiotic environment, the differences in abiotic microbiomes resulting from the various geographic locations of the institutions (in the mid-Atlantic, Midwest, and Southwest) may influence the types of microbes colonizing primate hair.

**(iv) Host body site.** When accounting for other variables, we found a strong effect of body site on microbiome diversity. Although, this finding was driven largely by differences between the tail (and to some extent the crown and thigh) and other body sites (see Fig. S3, Table S6, and Table S7 in the supplemental material). One explanation for the distinctness of the tail may be differences in its grooming traffic compared with other body sites. For example, the tail may have reduced grooming traffic from conspecifics, which has been demonstrated in Verreaux's sifaka (53), resulting in a unique microbiome structure compared with that of more heavily frequented body sites. Alternatively, the distinctness of tail hair microbiome structure may also be due to its regular contact with gut microbes in feces due to its proximity to the anus.

Our results are in stark contrast to those of humans, who exhibit substantial differences in skin/hair microbiome diversity across body sites. Much of the across-body site variation in the human microbiome is likely the result of microenvironmental variation across sites (29). For example, skin microbial communities sampled from different sites may be dominated by completely different families, as follows: the nares by *Corynebacteriaceae* and other *Actinobacteria*, the plantar heel by *Staphylococcaceae*, and the volar forearm by various *Proteobacteria* (29) (as shown in Fig. 3). The high variability in composition across multiple body sites in humans contrasts with our findings in nonhuman primates and suggests

that the more uniform distribution of relatively thick, long hair—as well as less variation in gland types (38)—across the bodies of nonhuman primates has a homogenizing effect on microbial communities (see Fig. S4 online at https://figshare.com/articles/figure/Hair_micro biome_diversity_within_and_across_primate_species/19860025).

Our results provide not only insight into primate variation but also a comparative context for understanding human evolution and uniqueness. The evolution of reduced body hair and the increase in eccrine gland density in the human lineage (38) has likely played a major role in differentiating the human skin and hair microbiome from that of other primates. This information in turn illustrates how an evolutionary change in one trait (distribution of body hair) can have an substantial impact on other key biological differences (microbiome diversity).

## MATERIALS AND METHODS

**Sample collection and DNA extraction.** Hair samples from captive primates housed at three U.S. institutions (Duke Lemur Center [Duke], Ft. Wayne Children's Zoo [FW], and Gladys Porter Zoo [GPZ]) were plucked by institution staff between 2006 and 2011. Most hairs were associated with their roots based on visual inspection, although we did not explicitly quantify this trait. The length of the hair shaft immersed in each collection tube was around or under 3 cm. All samples were collected with IACUC approval from Yale University (number 2010-11410) and the respective institutions where the primates were housed. We do not have information about the specific conditions under which the primates were housed (e.g., social group composition, diet, hormone cycling data, and bathing information), although we did request institution staff take samples only from healthy, adult individuals.

Our data set included a total of 158 hair samples representing primate species from each major clade (Catarrhini, Platyrrhini, and Strepsirhini). We obtained hair samples from up to 8 body sites (arm, back, belly, cheek, crown, thigh, proximal tail, and distal tail) from 21 individuals representing 12 genera/species (members of the same genus are of the same species) (see Table S1 in the supplemental material). Hair samples were stored in RNAlater at −80°C until DNA extraction. Our sampling included hair from one male and one female for all genera except for *Allenopithecus nigroviridis*, *Lemur catta*, *Mirza coquereli*, *Trachypithecus obscurus* (female-only), and *Cercopithecus neglectus* (male-only).

In preparation for DNA extraction, we cut hairs protruding from collection tubes with scissors cleaned with DNA Away (Thermofisher) and 70% ethanol as outlined in Tridico et al. (12) to prevent contamination. We extracted DNA using the Invitrogen PureLink microbiome DNA purification kit according to the manufacturer's protocol with the following modifications: (i) hair samples were moved between collection tubes via tweezers sterilized with DNA Away and 70% ethanol between each relocation, (ii) samples were incubated at 95°C for 10 min, (iii) samples in bead beater tubes were vortexed for 7 min both horizontally and vertically for a total of 14 min, and (iv) we set our centrifuge to its maximum speed of $12,100 \times g$ rather than $14,000 \times g$ that was recommended. The protocol modifications resulted from our prior experiments using different extraction kits and modifications to maximize DNA yield from hair samples. We quantified extracted DNA using a Qubit 3.0 fluorometer.

**Library preparation and sequencing.** We performed PCR amplification in triplicate using 515F-806R primers to amplify the V4 hypervariable region of the 16S rRNA gene (54). PCR cleanup was conducted using the Qiagen QIAquick PCR purification kit. Library preparation was completed according to the protocol specified in the Earth Microbiome Project and was sequenced in one run on the Illumina MiSeq platform with V3 chemistry and 201-bp read lengths (54–56) at the UMass Genomics Resource Laboratory. We generated a total of 49,245,096 raw reads with 93.39% of the reads associated with a Q value of >30. We specified a read depth of 20,000 reads and only included samples with at least 20,000 reads in downstream analyses, resulting in the exclusion of 1 hair sample (and 3 negatives). The average number of reads for our remaining 158 samples was 104,031, ranging from 25,635 to 219,861 reads.

FastQ files were imported into the QIIME2 pipeline for bioinformatic analyses (57). We used the DADA2 (58) plugin to identify amplicon sequence variants (ASVs) and to correct and/or remove sequencing errors, chimeric sequences, and chloroplast- and mitochondrion-associated ASVs. We used the naive Bayesian classifier method trained on SILVA (release 132) (59) reference sequences clustered at 99% similarity for taxonomic assignment of the ASVs.

**Alpha and beta diversity metrics.** We assessed alpha diversity using several metrics (Table 1 and 2) (60–63). We reported the relative abundances of microbial taxa at the phylum and genus level (Fig. 1 and Fig. S1). We used a series of Kruskal-Wallis tests with *post hoc* pairwise comparisons to examine differences in alpha diversity across body sites, institutions, sexes, and species (64). *Post hoc* test *P* values were adjusted for multiple comparisons using the false discovery rate (FDR) (65). We quantified beta diversity using weighted and unweighted UniFrac distances (66) and the Bray-Curtis dissimilarity index (67) and then visualized them via principal-coordinate analysis (PCoA). We used the UniFrac distances in PERMANOVA tests (68) to examine differences in microbiome composition among body sites, institutions, sexes, and species. We used 999 permutations for all PERMANOVAs. In addition, we used the PCoA scores of the first two axes as dependent variables in linear models. All alpha and beta diversity analyses were performed in QIIME2 (57). All figures were made in R (4.0.3) (69) and R Studio version 1.3.1073 (70) with the *ggplot2* (71), *tidyverse* (72), *ggh4x* (73), *qiime2R* (74), and *pals* (75) packages, except for Fig. S3, which was produced in QIIME2.

**Linear models.** Kruskal-Wallis and PERMANOVA analyses are employed commonly in microbiome analyses. However, they do not account for the potential covariation among predictor variables. Therefore, we used linear models (LMs) to examine alpha and beta diversity metrics. These models have been applied commonly to community ecology data (76, 77), including those focused on mammalian microbiome diversity (78). We conducted these analyses in R (4.0.3) (69) and R Studio version 1.3.1073 (70) using the lm function. We included four predictor variables: species identity, sex, body site, and institution. We examined the standardized residuals and Cook's distances in the full models to detect overly influential data points and check the model's assumptions. We used corrected Akaike's information criterion (AICc) to determine which combination of predictors best explained our dependent variables (79). The model with the lowest AICc value was considered the best, while models within 2 AICc values of the best model were considered equally good (79). We used the sum of AICc weights to determine which predictors best explained our data. Higher values indicate a better ability to predict the dependent variable (79). When examining beta diversity, we used PCoA scores from axis 1 and 2 as dependent variables. We conducted *post hoc* comparisons of alpha diversity and beta diversity metrics for the species identity and body site predictor variables. We did these comparisons by switching the reference category (set as the intercept) to obtain *P* values for pairwise differences between species and body sites. Then, we used the Benjamini-Hochberg FDR method to adjust *P* values for multiple comparisons (65). We used the *MuMIn* (80), *lme4* (81), and *lmerTest* (82) packages to calculate AICc values and sum of AICc weights.

We considered using phylogenetic models to analyze our data since multiple species are represented (83). However, given that a main goal of our study was to identify if there are species-level differences in microbiome diversity, using a phylogenetic model would not be helpful since variation due to interspecific differences would be accounted for by the model itself, resulting in a reduced ability to detect species-level effects. In addition, we considered using a linear mixed model with Individual identity (ID) set as the random effect because we obtained multiple samples from each individual. However, our data set includes samples from species represented by a single individual or one male and one female. Therefore, the inclusion of sex and species in our linear models nearly perfectly accounts for individual ID.

## SUPPLEMENTAL MATERIAL

Supplemental material is available online only.
**FIG S1**, JPG file, 2.3 MB.
**FIG S2**, JPG file, 0.5 MB.
**FIG S3**, JPG file, 0.9 MB.
**TABLE S1**, PDF file, 0.1 MB.
**TABLE S2**, PDF file, 0.1 MB.
**TABLE S3**, PDF file, 0.1 MB.
**TABLE S4**, PDF file, 0.1 MB.
**TABLE S5**, PDF file, 0.04 MB.
**TABLE S6**, PDF file, 0.05 MB.
**TABLE S7**, PDF file, 0.04 MB.

## ACKNOWLEDGMENTS

We thank the Gladys Porter Zoo, the Duke Lemur Center (Erin Ehmke and Sarah Zehr), and the Ft. Wayne Children's Zoo (Jennifer Diehl and Joe Smith) for providing hair samples. Sara Gutierrez assisted with data entry and management.

Funding for this study was provided generously by the Leakey Foundation, The National Science Foundation (BCS-1355021 and BCS-1606360), UMass Amherst, and The George Washington University.

This work was completed in part with resources provided by the University of Massachusetts Green High Performance Computing Cluster (GHPCC).

We greatly appreciate helpful comments from Associate Editor Sarah Hird and two anonymous reviewers on earlier versions of the manuscript.

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
