## [Reviewer comments · mSystems]

Hair microbiome diversity within and across primate species

Catherine Kitrinós, Rachel Bell, Brenda Bradley, and Jason Kamlar

Corresponding Author(s): Jason Kamlar, University of Massachusetts Amherst

Review Timeline:

Submission Date:	May 26, 2022
Editorial Decision:	June 28, 2022
Revision Received:	July 1, 2022
Accepted:	July 5, 2022

Editor: Sarah Hird

Reviewer(s): The reviewers have opted to remain anonymous.

Transaction Report:

DOI: <https://doi.org/10.1128/msystems.00478-22>

Hi Jason -

Your paper looks excellent and both re-reviewers agreed it was essentially ready to go. I am sending it back as a "minor modification" though, because both reviewers had a couple of very minor edits that I wanted to give you the opportunity to accept or reject as you saw fit. I don't need to see a comprehensive 'Response to Reviewers', just change what you'd like and upload the final version. If mSystems makes you upload a Response to Reviewers document, it's ok for it to be very bare bones. And this probably goes without saying, but it won't go out for review again.

Thanks for the cool paper and let me know if you have any questions.

I hope you're well!

Sarah
(form letter below)

June 28, 2022

Prof. Jason M. Kamilar
University of Massachusetts Amherst
Department of Anthropology
240 Hicks Way
Amherst, MA 01003

Re: mSystems00478-22 (Hair microbiome diversity within and across primate species)

Dear Prof. Jason M. Kamilar:

Thank you for submitting your manuscript to mSystems. We have completed our review and I am pleased to inform you that, in principle, we expect to accept it for publication in mSystems. However, acceptance will not be final until you have adequately addressed the reviewer comments.

Preparing Revision Guidelines

Sincerely,

Sarah Hird

Editor, mSystems

Journals Department
Reviewer comments:

Reviewer #1 (Comments for the Author):

This manuscript, especially with the modifications made in response to the reviewers' comments, presents an important contribution to the field of non-human primate microbiota and I commend the authors for using previously sampled individuals to complete this work. I am satisfied with the responses to the suggestions I presented. In particular, I appreciate the very thorough Supplemental Table 1-it helps to clarify the sampling scheme and draws attention to the breadth of this study, which is very important. If the authors feel there is room in the main manuscript, I might suggest making that Table non-supplementary.

I have only very minor comments remaining that I would suggest they address.

Line 63-64: This sentence feels a bit out of place without further explanation. I would suggest removing it OR adding a bit more explanation, especially because the links between hair microbiota and cancer are not a central theme of this study.

Lines 103-114: This is a very thorough and easy-to-follow paragraph contextualising sex differences and helps support the prediction you have about sex and hair microbiota.

Line 156: I would remove "we assumed that" from this sentence.

Line 262: *Cebus capucinus* is now *Cebus imitator*. I would suggest making note of this in a parenthetical in this sentence.

Lines 381-383: This sentence suggests two opposing outcomes which makes the next sentence about interpreting your results a little less clear. Please clarify the sentence to connect more directly with the results you found.

Line 390: Cool! I look forward to a follow-up about this topic!

Reviewer #2 (Comments for the Author):

The manuscript is substantially improved. The minor suggestions below will help clarify several points.

Line 64: Please clarify the statement by including "parasites and a human breast cancer cell line (26)."

Line 475: Consider replacing "irreversible" with significant or substantial. While it is likely that distribution of body hair had a significant impact on microbiome diversity, it is likely not irreversible since many factors impact microbiome diversity.

In some places taxonomic primate names are italicized (Fig 2a) and other places they are not (Fig 3a). Please be consistent.

It will be helpful for the reader if the primate clade is included after each primate host name in Figure 3 and supplemental figures. For example, for Species, designate "*Allenopithecus* (C)" and "*Cebus* (P)" etc.

Comments to Authors:

This manuscript, especially with the modifications made in response to the reviewers' comments, presents an important contribution to the field of non-human primate microbiota and I commend the authors for using previously sampled individuals to complete this work. I am satisfied with the responses to the suggestions I presented. In particular, I appreciate the very thorough Supplemental Table 1—it helps to clarify the sampling scheme and draws attention to the breadth of this study, which is very important. If the authors feel there is room in the main manuscript, I might suggest making that Table non-supplementary.

I have only very minor comments remaining that I would suggest they address.

Comments to Editor:

This work is certainly appropriate for mSystems. The supplemental material is important. I recommend that Table S1 be moved into the main manuscript, if space allows.

The authors did a great job addressing my comments. I believe this manuscript will be ready for publication after some final minor suggestions. This is an important topic and I commend the authors for their creativity and solid science.

Line 63-64: This sentence feels a bit out of place without further explanation. I would suggest removing it OR adding a bit more explanation, especially because the links between hair microbiota and cancer are not a central theme of this study.

Lines 103-114: This is a very thorough and easy-to-follow paragraph contextualising sex differences and helps support the prediction you have about sex and hair microbiota.

Line 156: I would remove “we assumed that” from this sentence.

Line 262: *Cebus capucinus* is now *Cebus imitator*. I would suggest making note of this in a parenthetical in this sentence.

Lines 381-383: This sentence suggests two opposing outcomes which makes the next sentence about interpreting your results a little less clear. Please clarify the sentence to connect more directly with the results you found.

Line 390: Cool! I look forward to a follow-up about this topic!

May 24, 2022

Dr. Rob Knight
University of California, San Diego
Senior Editor
mSystems

Dear Dr. Knight:

It is our pleasure to re-submit our manuscript “Hair and skin microbiome diversity within and across primate species”, for your consideration in *mSystems*. No version of this manuscript has been or is being submitted to any other journal. The initial version of our paper (mSystems00056-22) was rejected but received positive comments from two reviewers and the handling editor, Sarah Hird. After a busy spring semester, we are excited to submit our revised manuscript. Please find a detailed reply to the reviewers’ comments at the bottom of this letter. Our responses are in blue font.

Commensal microbes have been shown to play an extensive role in host health and behavior. Although integument-associated microbes play a key role in immunity for human hosts, most of our knowledge of the nonhuman primate microbiome comes from studies of the gut microbiome. Whereas the skin is the outermost protective organ for humans, in nonhuman primates the hair acts as the first physical line of defense in addition to being a platform for social interactions, and subsequently pathogen transmission. Given the role of hair in sociality and pathogen transmission, and the role of the microbiome in host health, it is possible a better understanding of hair-associated microbes will provide a greater understanding of primate host health and conservation. In this study we aimed to both characterize the microbial diversity present in the nonhuman primate hair microbiome and determine the factors predicting it.

We created the most comprehensive primate hair and skin dataset to date, including data from 12 nonhuman primate species sampled from eight body regions each. We find that the nonhuman primate hair microbiome is distinct from the human hair and skin microbiomes in that it is relatively uniform—as opposed to distinct—across body regions and is most abundant in gut-, environment-, and respiratory-associated microbiota rather than human skin-associated microbiota. Furthermore, we found that the nonhuman primate hair microbiome varies with host species identity, host sex, host environment, and host body site, with host species identity being the strongest predictor. This demonstrates that nonhuman primate hair microbiome diversity varies with both evolutionary and environmental factors and within and across primate species. These findings have important implications for understanding the biology and conservation of wild and captive primates, and the uniqueness of the human microbiome.

Respectfully,

Catherine Kitrinos (University of Massachusetts Amherst)
Rachel Bell (University of Massachusetts Amherst)
Brenda Bradley (The George Washington University)
Jason Kamilar (University of Massachusetts Amherst)

Our replies are in blue font

Reviewer comments:

Reviewer #1 (Comments for the Author):

Comments to Authors: This study seeks to compare and contrast skin and hair microbial communities across a wide variety of primate taxa. Hair samples plucked from captive non-human primates were compared with standard microbial diversity/richness metrics, using established methods. The authors found support for sex, body site, species, and institution predicting a variety of diversity metrics. Overall this paper is well-written, the methods are generally easy to follow (though the sampling scheme is unclear), and the interpretations of the data are not over-reaching (as sometimes can be the case in 16S-based projects). However, I find that there are several results that should be interpreted to a greater extent, and that more attention should be given the biology/ecology of the primate species that were included (see specific recommendations below). Further, there are additional issues that should be addressed.

Review

Major Comments

Line 123: Table 1 is referenced here, but the Table 1 that appears in the submission is microbiome diversity metrics. Please include a table that clearly outlines your sample size, sample species, institution where samples were collected, number of individuals, sex of individuals, body site, and (if you have this information) group size and co-housing of individuals at these institutions. In some zoos, multiple species are co-housed. If this was the case for any of your samples, it should be clarified in the text & table. Please also include data (or at least a general description) of grooming practices for each species in the text. This has implications for your predictions as they relate to lack of differences between body sites, as well as why humans might differ.

Thank you for this comment. We now include a table (Supplementary Table 1) with the requested information.

Line 165: Please include individual ID as a random effect in all models (or, if you did this, please make it clear in the text).

See below

Line 165: If individuals were sampled more than once, please include individual sample ID as a random effect in all linear models (or, if you did this, please make it clear in the text).

Thank you for this suggestion about Individual ID/random effects. We had the same idea at the beginning of the study. Though, the nature of our dataset and model construction nearly perfectly accounts for individual ID without explicitly incorporating this variable into the model. All but two species in our dataset are represented by one sex or one male and one female. Two species in our dataset (*Allenopithecus* and *Trachypithecus*) are represented by two females each. Therefore, these data in combination with including species and sex as predictors in our models accounts for 19 of the 21 individuals in our dataset. Since we are interested in the effects of species and sex differences on microbiome diversity we chose to keep these variables as predictors and not use Individual ID as a random effect. We now explain this in our methods section.

Line 308: These results are so cool! Can you please comment more directly about why you think the variation exists. Based on Table 1, there appears to be a huge range of mean ASVs recovered from these samples. You touch on olfactory reliance for *L. catta*, but what about some of the other notable results (e.g. *P. pithecia* is low, *T. obscurus* is high etc.)?

Thank you, we have added information about why we suspect certain clade level patterns in alpha diversity between these clades.

Line 361: I think it is worth also addressing hygiene practices (shaving, bathing frequently, use of antibacterial soaps etc) of humans here, and is worth including a bit more literature on human body site microbiota. Certainly the reduced hair on humans plays a role in beta diversity of human body sites, but I think that behavioral habits of humans are also certainly at play here and should be addressed in the discussion. I think it is also worth explaining potential differences between wild/captive primates. Are zoo animals ever groomed routinely by keepers? Are they given medications (topical or otherwise) that might influence skin/hair microbiota?

Thank you, we have added some information about human hygiene between lines 79 and 89.

Literature from human skin microbiome studies (which we discuss in the introduction) suggest that differences in microbiome diversity across body sites are due to topographical variation in the skin and differences in the distribution of certain skin features such as sweat glands. The evidence for a significant effect from washing is limited to the best of our knowledge.

Unfortunately, we do not have information about the grooming practices of the primates in captivity. Though, we have no reason to believe that the animals were groomed, or even handled regularly, by keepers. Medication influencing our results is also unlikely as we requested samples from only healthy individuals.

Minor Comments

Line 90: Please clarify the "dry" or "moist" inside the parenthetical. For readers unfamiliar with skin research, this sentence is a bit confusing.

Thank you, we have clarified these terms in the introduction.

Line 111: Please include a justification for the prediction that skin/hair microbiome diversity will differ across sexes (also, I believe "between" sexes may be more specific terminology here).

We now provide additional rationale for the possible differences between sexes.

Line 125: Please include full species names.

Thank you, we have included this in Supplementary Table 1

Line 131: Please explain why samples were incubated for 10 minutes at 90C and why they underwent bead beating for 7 minutes.

Thank you. This modification in the protocol yielded more DNA based on test samples. We now include this information in the paper.

Lines 221-225: Please include the other relevant statistics in-line here instead of just p-values (which are not particularly informative in the absence of other results of your tests).

Thank you, other test statistics have been added.

Line 280: The citation here is a bit confusing. Please make it clear that the citation is Clayton et al. As it stands, it sort of reads like 2 studies have found this same result.

Thank you, the phrasing has been modified.

Line 325-326: Could this also result from grooming networks that might vary with hormonal cycling? Do you have any repeat sampling from females in different stages of their reproductive cycle to test whether there are time when they are more/less different from males?

We do not have information about hormone cycling in this study but agree it would be an interesting factor to include in future studies. Also, individuals were sampled at only one time point.

Line 602. Figure 1. The remainder category is a bit odd and unspecific. Please explain what the Remainder category contains. The caption states that these are organized by host species, but only the genus name is included. Were these all from the same species? Please include species names in the figure and italicize them. Also, please change the y axis label to 'relative abundance'.

Thank you, information about the 'remainder' category has been added, as well as information on the primate species (all individuals of the same genus were of the same species). The labels have been italicized and the y axis has been changed.

Line 627: Please overlay the individual data points in this figure (I recommend geom_jitter for this) as it gives us a better understanding of sampling density and spread here. Again make it clear if these are all from the same species. Please italicize the genus name. Please label the y-axes on both figures.

Thank you, information about the species has been added, geom_jitter was used, genus names are now italicized and the y axis labels were added.

Line 635: Figure 3. The image quality is quite poor and blurry and it's very difficult to differentiate the clade lines. Same comment above about whether these are members of the same species, or different species in the same genus.

We added labels to the clade lines and have changed the resolution of the image.

Please ensure that all figures are colorblind friendly.

We appreciate this suggestion. Figures have been checked and changed to colorblind friendly palettes.

Supplementary Figure 2: I know it's a pain to deal with so many colors, but the reds are quite similar and very hard to differentiate in the plot. For example, the largest red section look like they are either "Other", Variovorax, Phascolarctobacterium. Please make it clearer which sections are which. One thing I have seen work well is organizing the phyla by relative abundance in the plot and legend, instead of alphabetically.

Thank you, we have changed this figure to resemble Figure 1 with level 6 taxa. The colors should be more differentiated now.

Reviewer #2 (Comments for the Author):

Kitrinos et al. investigate the hair (and associated hair bulb/root) microbiome from numerous non-human primates to assess the impact of environment and evolutionary history on microbe composition. Samples were collected across three different primate clades from three different institutions. Amplicon sequencing was done using the 16S V4 region to determine the microbe composition, and the authors assessed factors affecting the microbiota between species and individuals. In relation to the gut microbiota, there is not as much known about hair microbiota (especially across non-human primates) so this research adds important information to our body of knowledge in this field. As such, this should appeal to mSystems readers. Overall, the reviewer is enthusiastic about this research and appreciates the breadth of sampling across primate species. However, numerous aspects need further clarification, as indicated below. Of concern is the confusion between describing the study as a hair microbiome versus hair and skin microbiome study. It would be best discussed as a hair microbiome diversity analysis that may provide insight into the associated skin regions. Addressing the below points will improve the manuscript.

We are glad the reviewer is enthusiastic about this study. We provide detailed responses below.

1. The title is misleading by suggesting that both hair and, separately, skin microbiome diversity are assessed in the manuscript. It would be more accurate to title this "Hair microbiome diversity..." or "Hair shaft and hair root microbiome diversity..."

Thank you for this comment, we have changed the title and the contents to specify "hair" microbiome diversity.

2. In some places the authors state "hair microbiome diversity" (line 235, 257, etc.) and others they state "hair and skin microbiome diversity." This is confusing and is more accurate to state "hair microbiome diversity" since only hair was sampled. While the hair bulb/root is from within the follicle in the skin, and may have some skin cells attached, it would be fair to state that the hair microbiome diversity study may have connection to skin microbiome diversity, but the study should not be portrayed as a skin microbiome study since the authors did not separately sample skin. Therefore, line 25 in the abstract states "NHP hair and skin microbiome diversity" and would be more accurate as "hair microbiome diversity..." In many other places, the authors refer to their hair samples and hair microbiome diversity.

Thank you for this comment, we have changed the title and the contents to simply "hair" microbiome diversity.

Line 194: All hair samples were dominated by species from the Bacteria....

Line 235: Linear Models Predicting Hair Microbiome Diversity

Line 257: "We found that primate hair microbiome diversity..."

Thank you for this comment, we have changed the title and the contents to simply "hair" microbiome diversity.

Materials and Methods:

3. Please include a table (supplemental is sufficient) indicating which species and how many individuals were sampled at each institution. It is not clear if any two species were sampled from different institutions. Please include other relevant information such as age, sex, diet, body site sampled, etc. from each animal. Adding the common names of the primates in this table will allow all readers (regardless of their primate knowledge) to understand the context of the species chosen.

Thank you. We have now included a Supplementary Table 1 which includes sex, body site sampled, institution, species, individual name / ID, and date of sampling. We do not have information about diet. The individuals sampled were all adults.

4. The authors refer to Table 1 on lines 123-124, but Table 1 shows alpha diversity metrics and not animal details. It would be better to reference a table solely with animal information at this line.

Thank you for this comment, we have now included animal information in Supplementary Table 1.

5. Lines 115-117: The samples were collected at different geographic regions (Raleigh, NC; Ft. Wayne, IN; Brownsville, TX) that may have had different climates at the time of sampling. What month/season were samples collected? Were all animals housed in outdoor, indoor, or mixed enclosures? This is relevant to the environmental influences and would be useful to include in supplemental material.

Thank you, we have included information in Supplementary Table 1 about the month and year samples were collected. We do not have information about cohousing.

6. Lines 119-120: The plucked hair would have the hair bulb/root (likely not the follicle, which is the structure from where the hair grows out of the body). While it is reasonable to state that microbes on the hair are likely similar to those found on the associated skin, it should be clear to the reader that the authors sampled only hair and did not separately sample skin. Brinkac (ref 28) states "microbiota at each site mostly resembling microbial communities associated with adjacent cutaneous regions." The Brinkac study did not test the skin microbiota directly either, so caution should be used so as not to over-generalize without supporting data.

Thank you, we have now mentioned in lines 143-145 that we do not know if the roots were attached and have changed the title and contents to refer to the 'hair' microbiome.

7. Lines 119-120: Did all hairs have a full hair bulb/root? Did you assess the length of the hair shaft and the presence/size of the hair bulb/root prior to DNA extraction? Differing amounts of hair shaft vs. root/bulb could have disproportionately impacted the microbiome results and would be useful to report, if known.

Thank you, we have now mentioned in line 157 that we do not know if the roots were attached and have changed the title and contents to refer to the 'hair' microbiome. We do not know the length of the hair shaft though it is the subject of an ongoing project and most samples were around or under 3 cm so as to fit into the tube.

8. Were samples sequenced on one MiSeq run, or multiple? If multiple, were there any batch or sequencing run effects?

We have now mentioned they were completed on one MiSeq run in line 187.

9. Lines 142-143. Were the data rarefied, or filtered based on read depth? It is indicated that only samples with at least 12,000 reads were used. How many and which samples were excluded?

We have now included information about read filtering and the samples we excluded in lines 191 and 192.

Results:

10. In humans, age impacts microbiota composition. Did you assess animal age relative to aspects of microbial composition?

Thank you for this question. We did not assess animal age with respect to microbiome diversity as all of the individuals were adults. We do have future plans to examine microbiome diversity in response to different age classes.

11. Line 200: Please check order of figures and tables throughout. Supplementary Figure 2 is referenced before Supplementary Figure 1.

Thank you, the figures have been re-labeled accordingly.

12. Line 254: Where are Supplementary Tables 5-6? Please include.

Thank you, we have now added the tables.

Discussion:

13. Since Prevotella is a prominent microbe identified on hair, it would be worth noting that this is commonly found in the oral microbiome as well as gut. These results may reflect grooming practices of the primates in addition to captivity and dietary shifts.

Thank you, we have mentioned its presence in the human oral microbiome and added a citation.

Figures:

14. For all figures it would be more informative to group the species by clade instead of alphabetically. If there is a reason why they are better in alphabetical order, please explain.

Thank you, we have now grouped the primate host genera by clade where possible in stacked bar plots and box plots.

15. Figure 2: Please add p values, if any are significant. Otherwise, state that none are significantly different.

We have now added p values and other relevant statistics.

16. Fig 3: The dotted and dashed lines are difficult to distinguish on the figure (and supplemental figures as well). Please find another way to distinguish. Perhaps a long dashed line or a much thicker line would be easier to distinguish. Alternatively, the circles could be labeled with Platyrrhine and Strepsirrhine.

Instead of color-coding species alphabetically, it would help to group them by clade and choose colors that better distinguish species in the same clade. For example, it is very difficult to distinguish the greens of Eulemur and Hapalemur.

Thank you, we have changed the colors on these figures to be color blind friendly and easier to distinguish. We have added labels to the confidence intervals that specify clade.

July 5, 2022

Prof. Jason M. Kamilar
University of Massachusetts Amherst
Department of Anthropology
240 Hicks Way
Amherst, MA 01003

Re: mSystems00478-22R1 (Hair microbiome diversity within and across primate species)

Dear Prof. Jason M. Kamilar:

Your manuscript has been accepted, and I am forwarding it to the ASM Journals Department for publication. For your reference, ASM Journals' address is given below. Before it can be scheduled for publication, your manuscript will be checked by the mSystems production staff to make sure that all elements meet the technical requirements for publication. They will contact you if anything needs to be revised before copyediting and production can begin. Otherwise, you will be notified when your proofs are ready to be viewed.

Publication Fees:

If you would like to submit a potential Featured Image, please email a file and a short legend to mSystems@asmusa.org. Please note that we can only consider images that (i) the authors created or own and (ii) have not been previously published. By submitting, you agree that the image can be used under the same terms as the published article. File requirements: square dimensions (4" x 4"), 300 dpi resolution, RGB colorspace, TIF file format.

We recognize that the video files can become quite large, and so to avoid quality loss ASM suggests sending the video file via <https://www.wetransfer.com/>. When you have a final version of the video and the still ready to share, please send it to mSystems staff at mSystems@asmusa.org.

Sincerely,

Sarah Hird
Editor, mSystems

Journals Department
Table S4: Accept
Fig. S1.: Accept
Table S3: Accept
Table S2: Accept
Table S7: Accept
Fig. S3: Accept
Table S6: Accept
Fig. S2: Accept
Table S5: Accept
Table S1: Accept